# Reliable Methodologies and Impactful Tools to Control Fruit Tree Viruses

**Michel Ravelonandro**

UMR-BFP-1332-INRAE, Bordeaux, Bordeaux-University II, 71 Avenue Bourleaux,
33883 Villenave d'Ornon, France; michel.ravelonandro@wanadoo.fr; Tel.: +33-06-506-337-10

**Abstract:** Viruses are microbes that have high economic impacts on the ecosystem. Widely spread by humans, plant viruses infect not only crops but also wild species. There is neither a cure nor a treatment against viruses. While chemists have developed further research of inefficient curative products, the relevant concept based on sanitary measures is consistently valuable. In this context, two major strategies remain indisputable. First, there are control measures via diagnostics presently addressing the valuable technologies and tools developed in the last four decades. Second, there is the relevant use of modern biotechnology to improve the competitiveness of fruit-tree growers.

**Keywords:** fruit-trees; woody plants; molecular techniques; sanitary control; virus; disease; NGS; siRNA; CRISPR-Cas





## 1. Introduction

Control of viruses is a big concern because viral diseases are generally reported as the major cause of economic disaster concerning annual and perennial crops. Depending on the plant species, different areas from greenhouses, screenhouses and field orchards are used. The research studies focusing either on some academic improvement of science or the findings of solutions for the valuable exploitation of economic crops mainly contribute to controlling viral diseases [1]. A quarantine policy requiring local and international rules is requested for some models as for *Prunus* [2]. While crop species are independently considered according to the indigenous environment, the coupling of virus–crops is undoubtedly submitted to the natural interactions with insect vectors [3]. The consequences of these interactions exhibited either the growth perturbation or the adaptation of the plant materials [4].

As a reference, the European plant protection services via the European Food Safety Authority (EFSA) approved quarantine rules for *Prunus* that ordered labelling and plant certification before moving out and selling plants throughout the continent [2,4,5]. Basic work and a huge improvement of experimental protocols have been carried out related to the control of a quarantined disease, such as sharka [6]. Referring to the initial discoveries of sharka in the Balkan area, *plum pox virus* (PPV) adaptation and the geographical spread were updated [7]. As indicated previously, increased attention was paid to *Prunus* species, potential hosts, discrete reservoirs [8] and susceptible hosts [9]. Among the imperfect and disputable concepts were the contrasting observations between diseased and healthy *Prunus* trees that were split through serological tests [10].

Researchers developed several technologies to improve virus detection [10–12]. Virus infection in fruit trees is difficult to diagnose because these plants initially appeared either symptomless or showed suspicious symptoms. These doubtful scenarios that represent the initial steps for detection were interestingly explored. A field survey and the search of the possible insect vectors were the reliable measures preceding the leaf sampling, which was the real start [13]. Of innovative relevance to these observations, researchers brought sensing technology (global positioning systems, GPS) to trace the sources and follow up on the disease impacts in orchards [14]. In order to better understand the plant

virus epidemiology, both types of samples, including diseased and symptomless leaves, were collected.

The rationale for regulation was derived from grapevines, with symptoms as a positive control to explore trials to characterise the unknown viral cause [15]. The hypothesis suggested that laboratory studies will confirm the occurrence of a virus genome, which will resolve the doubt about the causative virus [16]. Field release opens any experimental fruit trees to an uncontrolled environment wherever several viruses can co-exist or threaten them [17]. Looking through the recent manuscripts published in this present decade, it is possible to discover the presence of either hidden or new viruses. Particularly, it was detected how difficult it has been to find out new viruses via metagenomic technologies, such as Next Generation Sequencing (NGS), High Throughput Sequencing (HTS) [18]. In addition, fruit tree viruses are known to present a latent form and uneven distribution in the host plant [12,19]. Managing fruit trees in Europe gave rise to an economic challenge. Conversely to growers in Northern Europe, who use the oldest cultivars of temperate fruit trees (*Prunus*, *Malus*, etc.), those in Southern Europe, having warmer climates, have recently introduced new tropical crops [20]. Obviously, there is some diversity of models to relevantly show that controlled measures of virus infection are a long-lasting and costly challenge.

In response to professional demand, including scientists, growers, breeders and regulators, this review provides an updated platform of approach to control fruit tree viruses. Laboratory analysis [21] can demystify whether fruit trees are fully diseased or if they can be deemed virus reservoirs [4,8,9]. Conversely, resistant clones are virus-free [22]. Because fruit trees grow in the field for years, the maintenance of diseased fruit trees in greenhouse conditions is a determinant to maintain their relevant phenotypes under natural conditions. In the same way, this is also a basic step to perform resistance assays before they are transferred to natural conditions. As proof of concept, the release of these resistant clones in an uncontrolled environment will meaningfully confirm the sustainable trait expectedly developed.

## 2. Fruit Tree Virus Disease Spread

As opposed to annual crops that readily show typical traces of insect bites in leaves and readily exhibit symptoms, perennial plants are less affected. Uncertain and doubtful hypotheses led epidemiologists to track and identify any possible vector for virus transmission [23]. The need to investigate possible sources of perennial plant viruses in the environment is substantial. Concerning sharka, all *Prunus* species in Europe have been tested, and most have exhibited PPV disease. While a few species are resistant, all the commercially exploited cultivars are susceptible. Virus transmissibility was closely related to aphid distribution [23,24]. In this regard, virus strain and aphid vector can form a dependent couple, e.g., in Spain and North America. While the two strains of PPV (D and M) that serologically differ can similarly cause symptoms in any susceptible cultivars [25,26], only a molecular investigation can arguably explain the causative plant phenotypes [27]. In most cases, Europeans took the observations about the high incidence of PPV that has so far caused the spread of sharka [4,9] from the Balkan areas to Europe. PPV strain C, isolated from cherry trees [28], is restrictively expanded in Ukraine and Russia [29]. While greenhouse and laboratory studies did not solve the particular properties of PPV-C (cherry strain) [30], uncertainties about the timing and the geographical distribution of these spotted areas in Ukraine and Russia remain elusive.

Interestingly, perennial plants stored in a quarantine service are considered somewhat virus-free collections. As an academic model, the Australian plant protection in Victoria provided a good example wherever it was essential to better understand virus genome spread in perennials. If fruit trees, diversely growing in the country, share a common infection, the label about safe commercial fruit trees is important. In the absence of potential vectors, *Malus* infected with the *apple stem grooving virus* (ASGV) raised some problems of infection. If the exact date of their introduction did not allow them to trust their sanitary

status, the recent studies conducted by Kinoti et al. [31] provided some evidence that the virus had been propagated by grafting. At a time when Europeans introduced mother trees to Australia, archives were the sole sources that could help to inform about their origins. The scenario evoking the *apple chlorotic leaf spot virus* (ACLSV) spread in Australia is a lesson about the impact of trade. Undoubtedly, long-distance transport of the virus came from unintentional human action. Obviously, the high spread of the *Prunus necrotic ringspot virus* (PNRSV) in fruit trees means that these serve as virus reservoirs. If several fruit trees were diseased, the successive rounds of cross-hybridisation could not be excluded because they could lead to virus dispersion via pollination.

Parallel to the exploitation of the *Prunus* species in Europe already cited during the Roman Empire, ancient China was also well-known as a hot spot of pomiculture. Recently, researchers in China clarified the occurrence of the *apple necrotic mosaic virus* (ApNMV) and revealed the areas where it occurred. For years, ApNMV has strongly impacted southwest China. If it is present in other locations in Asia, such as Japan and Korea, it has been introduced by humans [32]. Meanwhile, the recent discovery of the *Grapevine Pinot Gris Virus* (GPGV), a trichovirus initially detected in Italy in 2012 [16], reflects its first occurrence in Europe. For a trichovirus where the insect vector was not accurately identified, the subsequent investigation pointed out that GPGV is a virus widely present in five continents [16,33–38]. Undoubtedly, the propagation of clones by grafting was the most common way to spread GPGV in grapevines. Regardless of either cultivars or rootstocks, which have recently been identified as virus sources, molecular sequencing confirmed the large distribution of GPGV and indicated China as the centre of this virus spread [38].

### 3. Molecular Tools to Control Fruit Tree Viruses and Impactful High Throughput-Sequencing (HTS) Methodologies to Detect Hidden Fruit Tree Viruses

To improve virus detection, leaf sampling was the beginning, and total RNA was extracted from leaves. To avoid any difference in the expressed symptoms, a transcriptomic method based on the bulked system was adopted [39]. The expectation of virus infection prevailed in the sampled lot of field materials. Routinely applied to detect virus genome by RT-PCR [40], this strategy confirmed the occurrence of virus infection from four weeks after bud breaking. As indicated previously, sensing technology (GPS) is helping to trace the sources and the disease impact [41].

The grapevine, one of the perennial crops domesticated by humans, is among the relevant models widely explored in fruit trees. With some technical limitations due to the lack of adequate protocols for nucleic acid extraction, the development of new technologies based on metagenomic deep sequencing methodologies was laborious. Moreover, the successful improvement of these methodologies through the use of chaotropic solutions and magnetic beads led quickly to the isolation of pure nucleic acid. As a key factor for sequencing, these templates were crucial to the recent discovery of new viruses. With the powerful application of shotgun metagenomic sequencing (NGS, among others), access to bioinformatics resources sped up the identification process [42]. Disregarding RNA or DNA [43,44] viruses, the classical reliance on uncommon symptoms and the adequate transfer onto herbaceous hosts allowed the confirmation of the virus identification. Regardless of either cultivar or rootstocks, the various attempts to correlate any unknown symptoms on grapes were broadly successful. If the resulting sequencing reads permitted covering the whole viral genome that supported the large distribution of GPGV [38], some studies aiming to identify GPGV provided interesting data that eluded the occurrence of co-existing viruses, notably in the Czech Republic where the initial investigation was basically to characterize *Grapevine leafroll-associated virus-1*, GLRaV-1, *Grapevine Fleck virus*, GFlV, *Grapevine Vitis A*, GVA and *Grapevine Vitis B*, GVB [16]. Czech researchers met their needs because the data collected from nucleic acid sequencing led to the discovery of several RNA viruses, including *Grapevine rupestris stem pitting-associated virus*, GRSPaV, *Grapevine Syrah virus-1*, GSyV . . . and viroids (*Hop Stunt viroid*, HSVd and *Grapevine Yellow speckle viroid*, GYSV). These studies reliably identified new viruses and viroid in the grapevine.

The advancements performed on virus diagnosis require expanded research about the identification of relevant insect vectors involved in transmission. In order to contribute to the knowledge of grapevine viruses, a recent study with selected cultivars growing in different regions in France significantly provided a pool of molecular data of the GPGV genome that helped the study of the virus genome evolution [38].

As indicated previously, DNA viruses can occur in fruit trees [43,44]. Diseased grapevines growing in the Mediterranean basin led Israeli researchers to find the unprecedented geminivirus, *Grapevine geminivirus virus A,* GGVA [45]. The adopted approach started from nucleic acid extracts that were molecularly detected via rolling circle amplification (RCA) with the typical nonanucleotide sequence TAATATTAC followed by the HTS technology. Through pairwise study and multiple alignments, GGVA was identified. Molecular data confirmed the phylogenetic relationships with other geminiviridae taxa. While the geminivirus frequently infects herbaceous hosts, other new closely related geminiviruses infecting fruit trees were also identified, as had been for *grapevine red blotch virus*, GRBV [44], and *apple geminivirus*, AGmV [46]. As these viruses were discovered in different areas of the world, including Canada, China, Hungary, Israel, South Korea and the USA, there are no more available data about their evolutionary history. Similarly, the identification of the insect vector transmission needs to be elucidated.

As presently developed with grapevine, a widespread perennial crop domesticated by humans, controversial consideration with symptomless samples was also restrictively explored through RNA-Seq technologies [47]. Metagenomics-NGS was the basic approach for detecting pieces of the virus genome [48]. Obviously, RNA-Seq, commonly used to detect viral RNA in different parts of the entire plant, partially solved the uneven distribution in fruit trees. Several manuscripts have successfully reported the discovery of hidden virus genomes during this last decade [16,19,33,49]. The HTS approach allowed the discovery of the *Prunus virus T* [50] and *Prunus virus F* [51], two new viruses detected in an unbalanced presence in fruit trees. With regard to the fruit trees that did not show evidence of infection, only laboratory techniques with technologies based on sequencing can confirm whether a virus is widely spread or present in a low titre [52]. These studies relied on the occurrence of the *Plum bark necrosis stem pitting-associated virus* (PBSPaV) that spread by grafting and no known vector [53,54].

The reason for the purification of the mixed population is an attempt to regard the dominant virus *versus* the minor pest. Disregarding the age of the sampled tissues, the method based on the bulked system allowed the confirmation that the tested cultivar was either infected or virus-free. These successful examples deriving from the exploitation of the virus genome database involved the smart design of a panel of primer-probe sets [16]. Using bioinformatics, the design of different primers that specifically recognised the respective templates allowed differentiating the co-existing virus genomes [33]. Consequently, PCR based on multiple specific primers confirmed that molecular studies optimized for such analysis could help in problems of virus genome identification. The reliable use of RNA-Seq for the detection of unknown virus genomes is related to the different ratios of RNA between virus genomes co-existing in plants [16,33,54]. However, the diversity of genera of viral RNA detected in grapevines pointed out the apparent complexity of symptoms related to the interactions between viruses and perennial hosts. Huge progress has been achieved in the investigation of a group of viruses frequently present in plants. The process is based on the potential exploitation of molecular probes via bioinformatics [55,56].

## 4. Maintenance of Virus Collection

In contrast to grain crops represented by annual plants that require careful survey through either the initial apparition of insect vectors of a virus or the development of leaf symptoms, fruit trees are less economically considered in agriculture. This is because coloured and edible fruits, taken as references, substantially reflect an attractive and safe image attributed to high-quality fruits. Production of commercial fruits follows a long period (three seasons, from spring to fall) in a temperate climate. From the first fall, whenever trees

start the dormancy cycle to the second, growers maintain fruit trees. According to the cycle from pruning to fruit harvesting, growers and researchers remain closely involved and mainly responsible for trees. Following the surveillance, viral disease study from bark or young leaves should not be doubtful [40,42,54]. Understanding and predicting the effects of fruit tree viruses (e.g., PPV) are difficult and long-lasting events [10–12,22,40]. From the bud-breaking period to fruit ripening, the latent form of the viral disease can delay the appearance of symptoms. In the USA, the first discovery of PPV was delayed [3,4,8,11] because the typical forms of symptoms were only recognisable on fruits (several white rings on peaches, gummy and deformed plums and white rings on apricot stones) (Figure 1).

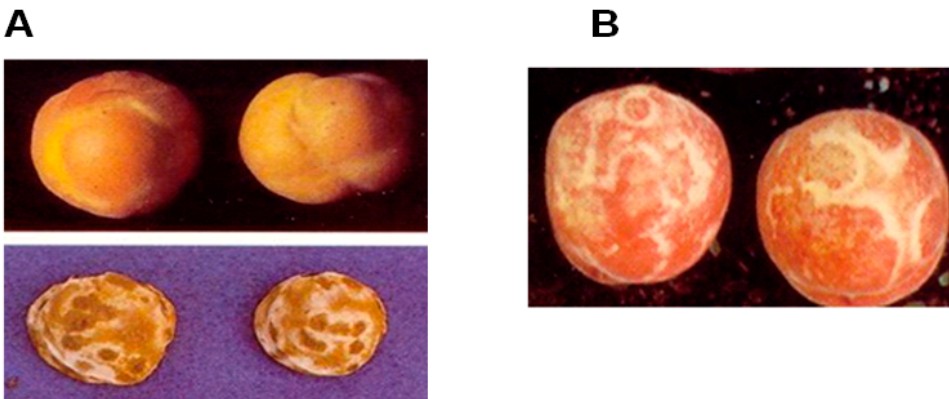

**Figure 1.** Typical symptoms on infected fruits: (**A**) apricots (deformed fruits and white rings on stones); (**B**) white circles on peach nectarine.

Over the number of fruit trees sampled, logistics is the determinant. How big are the greenhouses? Wherever infected plant materials (barks and shoots) are seasonally collected and used as virus sources, how safe are the high-containment greenhouses used to maintain plant material infected with the quarantine virus (e.g., PPV and viroids). Safe containment was designed to protect plants against any invasive vectors and, notably, uncontrolled atmospheric conditions. To study young shoots developed after a short period of the artificial dormancy cycle (two months), the first observations are conducted on the first expanded leaves. Symptom appearance is one of the major steps for disease identification.

To better study plant viruses, researchers should comply with ethical and environmental laws [2,3]. According to the sanitary status of the viruses (quarantine or not), before they are studied in the laboratory, they were domesticated either in a greenhouse (more expensive) or screenhouse (cheaper). Careful managing is required to maintain any virus strain collection [12]. In general, virus strain(s) can differ from others by their restricted storage in a natural host (e.g., PPV-C in cherry trees) or their variable impact in indexing hosts (*P. tomentosa* and *GF305* peach tree, mild symptoms with PPV-D and severe symptoms with PPV-M) [25,26]. Referring to the coupling of *Prunus*–PPV, the European plant protection services inherited helpful methods and protocols to improve the policy community [2]. Experimental studies showed that following the huge collection of PPV sampled from the field [12], domesticated viruses can be legally and safely stored in a high containment greenhouse. Disregarding RNA (PNRSV, PPV, PBNSPV and *Citrus psorosis virus* (CPsV) among others) or DNA (*grapevine geminivirus A*, GGVA, GRBV and AGmV) viruses, attempts to solely graft diseased buds to rootstocks allow the transfer of the new suspected virus. While symptoms appear after bud-breaking, this protocol, the way of virus passage in the indexing perennial host, confirms the high susceptibility of the propagated cultivar. Similar to the coupled plum–PPV interactions, these were also achieved with grapevine and citrus possibly infected with either viral RNA [57] or viroid [16].

### 5. Genetic Engineering of Viral RNAi in the Sustainability of Fruit Trees

In these last decades, serious viral diseases that affect fruit trees for years were challenged either with cross-protection (CPsV) [58] or the classical breeding (PPV) [30] techniques. The paucity of natural resistance sources and the inefficiency of both methods led to the alternate development of the genetic engineering approach. In line with huge progress in molecular virology, plant transformation of perennial crops provided a lot of transgenic plants used as challengers to viral diseases [59,60]. By engineering viral gene (intron hairpin capsid protein sequence, ihCP) constructs allowing specific production of RNAi, successful achievement was performed with transformed Prunus domestica [60] and sweet orange Citrus [61]. Propagated by grafting onto susceptible rootstocks, greenhouse studies on these perennial plants showed that the engineering of small interfering RNA (siRNA) is an effective technology ensuring the accurate and safe use of plant biotechnology. Virus gene interference [61–63] that requires the transcription of the viral dsRNA is well documented and understood. Sliced into siRNA, these siRNAs via RISC (RNA-induced silencing complex) trigger the targeted virus genome that is selectively degraded by the plant dicer system. The siRNA molecules play a determinant role in setting up a sustainable program for genetic resistance. These assays with virus gene interference reflected the ultimate step preceding the field releasing of plants [64]. Following the diversity of small-scale field tests of virus resistance, the technological progress of genetic engineering is correlatively great for fruit-tree improvement. To date, tools and technologies to compete with classical breeding techniques have clarified the molecular mechanism of how the use of either virus [61] or plant [65] genes in plants was genetically and efficiently implicated. Undoubtedly, the development of RNAi technology is the safest and faster alternative to any challenging techniques [50].

The conclusion is that genetic engineering with virus genes solved the sharka problem [3,4,7,8,30]. For more than two decades, we showed that the silencing displayed by '*HoneySweet*' is exhaustive. European growers can hope for a real challenge against sharka. Even if GM plants are valuable, it is still critical; however, the deregulation by the EFSA will offer an attractive opportunity to solve the sharka problem. For three substantial reasons: first, '*HoneySweet*' is a resistant clone; second, it harbours some inheritable gene; third, a bankable source of a breeding program. It means that plant biotechnology is the ultimate hope for growers.

### 6. Gene Editing: A New Technology for Challenging Fruit Tree Viruses

Gene editing is an additional technology for engineering a reliable guide RNA to knock out genetic information of the undesired virus. Inspired by the powerful RNAi technology, researchers increasingly developed molecular techniques to target any undesired pest that threatened crops. Clustered regularly interspaced short palindromic repeat (CRISPR) associated (cas) genes technology, CRISPR/Cas9 [66], which inspired molecular geneticists, was successfully used to create an engineered banana. Among the serious viral diseases affecting fruit trees are banana viruses, including *banana streak virus* (BSV) and *banana bunchy-top virus*, BBTV. Erroneously regarded as fruit trees, banana is the relevant model revealing the high yield loss caused by viruses. BSV harbours a reverse-transcribing genome with a circular double-stranded DNA genome that is replicated via a virus-encoded reverse transcriptase to produce a form integrated into the banana genome (endogenous paratretroviruses, EPRV) and an exogenous episomal form (eBSV) [67]. Although the latent form of the disease does not show any visible sign of infection, the occurrence of the disease is related to this eBSV, harboured mainly in *Musa balbisiana* (B genome). In response to abiotic or genomic stresses, both integrated viral sequences EPRV and eBSV can be reactivated and cause banana infection. Disregarding the routine spread via cuttings, BSV is naturally transmitted by mealybugs (*Planococcus*). Insect vectors form a serious threat for the banana–BSV pathosystem, notably its outbreak in commercial crops (hybrids). Targeting the eBSV by editing the virus sequence in Plantain (Musa spp., AAB genome) was the strategy developed to knock out the virus infection via CRISPR/Cas9 [68]. Attempts

were made to target and inactivate eBSV sequences through three mutations targeting the ORF1, ORF2 and ORF3 of the integrated sites via the CRISPR/Cas9 constructs. The designated Cas9 plasmid vector was used to transform embryogenic cells via *Agrobacterium tumefaciens* [69]. Regenerated plantlets were transferred and acclimatized in a greenhouse. Interestingly, the preliminary assessment of the regenerated banana showed that the CRISPR/Cas9 process developed with the banana was successful. Evidently, the benefits earned with the edited banana should be verified through multiple cross-hybridisation. Improved B genome would be the suitable banana for commercial development. The question that arose is the availability of a stable and durable resistance [68].

Even if the acceptability of gene-editing technology is similarly pending with GMO technology in Europe, the idea to interfere with the virus genome is a great challenge. Undeniably, the plant phenotype is unchanged, and the rationale for the engineered *Prunus* is to block the virus genome spread as indicated by the interference with the functional gene. This is the innovative approach to reliably tackle PPV upon the classical breeding program [30]. In order to control PPV, the development of an innovative gene-editing technology argues that the discovery of a cluster of meprin and TRAF-C homology domain (MATHd) containing-genes from apricot-trees is consistent with virus genome interference [70]. While silencing is basically attractive, its introduction in perennials should suggest that the virus genome interference could be expanded to other *Prunus* species.

## 7. Conclusions

Different results about controlled measures on fruit tree disease impacts were reported in this review. Fruit trees, which grow for years in the environment, represent a very sensitive case because the diagnosis of the virus disease is very difficult. While the identification of virus vectors is basically the required first step, there is a substantial delay, designated as latency, which is depicted through the absence of symptoms prior to their appearance. In this regard, epidemiology studies became not only complicated but randomly set. While an asymptomatic or latent form of the disease did not assist growers, laboratory studies based on the RNA-Seq and metagenomic approaches brought useful information. Without any cytopathic effects, the detection of known viruses with a low titre that can co-exist with unknown viral RNA frequently occurred. To defend the rational science in perennial crops, the study with the '*HoneySweet*' plum [22,30,38,39,42,43,60,62–64] has highlighted the prosperous technologies based on genetic engineering. Fighting wisely with gene-editing technology to control virus genomes would be a new challenge. If the edited banana develops a stable and durable resistance to BSV infection, the CRISPR/Cas13 strategy [71] that targets the viral RNA transcripts is a remarkable alternate to the engineered siRNA strategy already developed in *Prunus* and *Citrus*.

**Funding:** This research received no external funding.

**Conflicts of Interest:** The author declares no conflict of interest.

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
