# Peer review of "Reliable Methodologies and Impactful Tools to Control Fruit Tree Viruses"

_2673-7655, doi:10.3390/crops1010005_

Round 1
Reviewer 1 Report
Your review "Reliable methodologies and impactful tools to control fruit tree viruses" is interesting. However, you have a very verbose language style which sometimes distracts from the meaning of the sentence. Plain and concise writing is very important and gives unambiguous meaning to the understanding of what was being conveyed.
I have attached a PDF with comments for your review.
regards

Author Response
I would thank the reviewer 1 who had carefully read the manuscript, and brought useful and constructive comments, for improving the review. I incorporated all quested changes and addressed my corrections and responses in order to clarify either the doubtful or confused sentences. I am addressing here my thankful appreciation about the cooperative assistance.
I reported two versions of the manuscript, the one with red marks and the second one without mark.
Response to Reviewer 1 Comments
Point 1: Line 30: EFSA, spell out acronyms in the firt instance
Response 1: European Food Safety Authority (EFSA)
Point 2: Line 58; spell out acronym in the firt instance, i.e Next Generation Sequencing (NGS)
Response 2: Next Generation Sequencing (NGS)
Point 3: Line 58 delete and replace with (HTS)
Response 3: done
Point 4: Line 62; what does better mean? Just have as “having warmer climates…”
Response 4: better has been deleted
Point 5: Line 74: add a full stop
Response 5: done
Point 6: Line 114: replace Grey with Gris
Response 6: done
Point 7: Line 149: not sure what this means? “obtains an excellent diagnosis”?
Response 7: met their needs
Point 8: Line 152: capital V
Response 8: GYSV
Point 9: Line 153: delete relevantly
Response 9: done
Point 10: Line 154: change to “requires”
Response 10: requires
Point 11: Line 154: change to relevant insect vectors
Response 11: relevant insect vectors
Point 12: Line 183: replace with Plum bark necrosis stem pitting-associated virus (PBSPaV)
Response 12: Plum bark necrosis stem pitting-associated virus (PBSPaV)
Point 13: Line 193-194: replace with RNA-Seq
Response 13: RNA-Seq
Point 14: Line 209-211; not sure this is relevant
Response 14: Following the surveillance, viral disease study from bark or young leaves should not be doubtful
Point 15: Line 213-218: these sentences are disjointed and don’t make sense or flow together. please re-write
Response 15: From the bud-breaking period to fruit ripening, the latent form of the viral disease can delay the apparition of symptom.
Point 16: Line 250: Do you mean original
Response 16: propagated cultivar
Point 17: Line 251: replace with citrus as consistent with grapevine
Response 17: citrus
Point 18: Line 251: either or what?
Response 18: either viral RNA [58] or viroid [16, 33]
Point 19: Line 253; long what? periods, term, time?
Response 19: long years
Point 20: Line 260-261: keep consistent namimg convention plum Prunus (61) and sweet orange citrus (62):
Response 20: transformed Prunus domestica and sweet orange Citrus
Point 21: Line 262: spell out acronym in the first instance “engineering of small interfering RNA (siRNA)
Response 21: engineering of small interfering RNA (siRNA)
Point 22: Line 265: delete as you should have added it earlier
Response 22: siRNA
Point 23: Line 273: replace with were
Response 23: were
Point 24: Line 276; not sure the sharka problem is fixed entirely technology has gone a long way to controlling it but it is not eradicated
Response 24: has solved the sharka problem [3, 4, 7, 8, 30].
Point 25: Line 280: delete if acronym is spelled out earlier in the manuscript.
Response 25:EFSA
Point 26: Line 291: add “Virus”
Response 26: banana streak virus
Point 27: Line 293: replace with reverse-transcribing pararetroid is confusing
Response 27: reverse-transcribing genome
Point 28: Line 293: replace with “with a circular double-stranded DNA genome that is replicated via…”
Response 28: with a circular double-stranded DNA genome that is replicated via…”
Point 29: Line 298: replace “from” with “in”
Response 29: done
Point 30: Line 301: Planococcus spp.
Response 30: done
Point 31: Line 303: What does this stand for?.
Response 31: Plantain ( Musa spp ., AAB genome)
Point 32: Line 313: I don’t understand the premise of this?
Response 32: deleted
Point 33: Line 321: what does it mean?
Response 33: a cluster of meprin and TRAF-C homology domain (MATHd) containing - genes
Point 34: Line 333: “approaches”
Response 34: done
Point 35: Line 334: not sure what scenario is being revealed?
Response 35: deleted
Reviewer 2 Report
Review report on the review of M. Ravelonandro entitled: „Reliable methodologies and impactful tools to control fruit tree viruses”
According to the title this suggested review wanted to summarize reliable methodologies to control fruit tree viruses.
This is a very important question and would deserve a comprehensive review.
Although this suggested manuscript collected several important papers, I think that in this present form it is not suitable for publication.
Although from the title we think that it will give us information about fruit trees, most information is collected from grapevine. The main flow of the paper is very distracted, for a review a more careful data collection would need.
There are several different reviews about citrus, apple and Prunus infecting viruses and their control, if a new review is written it would need and extra information, which I miss from this manuscript.
Because of these serious shortages I don’t suggest to accept this review in MDPI Crops.
Author Response
- First I would thank the reviewer2nd for having commented and amended his or her critics about the review. However I do not understand your restrictive comments considering the manuscript as shortage about either grapevine crops or the three fruit crops including Malus, Citrus and Prunus. There is a key-point that I recognize, fruit crops are not similar to annual plants. Because first, they required specific areas including greenhouse, screenhouse and experimental field about their use, secondly they are labelled as quarantine pests so their restrictive use (tests and sampling) should follow a strict regulation in a high containment greenhouse. And the last but not the least, fruit tree viruses are among the current bio-aggressor, difficult to detect. This review reports data from different studies achieved in both high containment and natural controlled conditions. Upon the economic role of fruit trees in temperate countries, determinant choice and priority should be taken in order to highlight the important findings reported in this review. It is a pity that you found it as a very restrictive review.
-
Response to Reviewer 2 Comments
Point 1: Although from the title we think that it will give us information about fruit trees, most information is collected from grapevine. The main flow of the paper is very distracted, for a review a more careful data collection would need.
Response 1: This review is not restrictive. It consisted of substantial results dealing either diagnostics or virus control about five major fruit crops including banana, Citrus, Malus, Prunus and Vitis. I developed some knowledgeable models in the manuscript. I focused on the significant findings based onto the molecular strategy used to detect either the hidden or the coexisting virus genomes (detection of viroids, GPGV in Czech Republic and the different French vineyards..PPV, Ilarvirus….). In the last section of the manuscript, I expanded the review in the development of the modern biotechnology including the siRNA (Prunus and Citrus) and the CRISPR-cas9 (banana) technologies. Both techniques as alternate to the classical breeding approach match to the technological transfer to end-users (fruit growers)..
Point 2: There are several different reviews about citrus, apple and Prunus infecting viruses and their control, if a new review is written it would need and extra information, which I miss from this manuscript. Because of these serious shortages
Response 2: As innovative ways, I provided an accurate and technical guidance (from suspicious symptoms in fruit trees to molecular sequencing, NGS and HTS), on coexisting virus in economic fruit trees like grapevine, apple and Prunus. While the environmental impacts of insect vectors remained unsolved in the control of these fruit crop models, moreover the geographical distribution of virus (Ilarvirus, GPGV…) related with trade remains problematic, notably for quarantine virus like PPV, to reshape the international regulatory policies.
- Enclosed is a revised version with marks.
Reviewer 3 Report
This is a nice review manuscript on the recent information of fruit tree virus detection and control methodologies and tools. The manuscript was well organized and contained recent and useful information which can be applicable to the fields.
Author Response
I would like to thank the reviewer 3 about his (or her) relevant appreciation of the manuscript to be acceptable as it is.
Response to Reviewer 3 Comments
Point 1: This is a nice review manuscript on the recent information of fruit tree virus detection and control methodologies and tools. The manuscript was well organized and contained recent and useful information which can be applicable to the fields
Response 1: Thank you very much, I appreciate.
Emclosed is a modified version of the manuscript
Round 2
Reviewer 1 Report
Thank for addressing the corrections, I have attached manuscript with some minor corrections. Well done on your hard work.

Author Response
- Reviewer1:
- I agree with these following points:
- 213: Appearance of symptoms
- L248: Deletion of long
- However:
- I) l.271 ‘helped to solve’ : In our concept of the Honeysweet plum resistant to PPV infection, plant biotechnology is an alternate solution to the classical breeding technique. I absolutely reject the suggested use of “helped to solve”because this transgenic plant shows a stable and durable resistance in field natural conditions. Confidentially this clone is deregulated in the USA, and, in Europe, we are preparing a regulatory dossier for the EFSA (European Union). I am so sorry that I insist here, that plant biotechnology came up with the scientific approach and not any approximate methods. The use of “solved’ is correct.
- Ii) l.289 change to virus. BSV is a reverse-transcribing genome with a circular double-stranded DNA genome that is replicated… was changed to BSV harbors a reverse-transcribing genome with a circular double-stranded DNA genome that is replicated I am strongly consistent on the accurate use of “genome” because you contested the preceding use of a pararetroid virus about BSV, so I have included “harbors”instead of “is”.

Reviewer 2 Report
I think that careful revision of the manuscript mainly according to the reviewer1 opinion improved its quality.
I do share the opinion of the author that viruses of grapevine and fruit trees have resemble and needs similar protection strategies. My remark aimed to suggest a rephrasing of the title, which could include grapevine also, or woody perennial plants instead of fruit trees. I still would suggest this possibility.
Otherwise, I thank that in this revised form the manuscript can be accepted for publication in MDPI crops.
Author Response
- Reviewer2:
- To make a change to the title is very awkward because the use of “fruit trees” avoid any inclusion of “papaya” that is the known example of successful fruit crop in the fight against papaya ringspot virus (PRSV). For two reasons, first, papaya is a herbaceous plant and second there are many literature reviews about the papaya/PRSV couple. The word of “perennial” means the restrictive development with plum-trees, grapevine, apple and citrus. None of these perennials can be controlled with the innovative CRISPR-Cas technology, excepted banana, that I indicated in the manuscript, as an erroneous model, because it is a monocotyledon plant. For making short, the use of“fruit trees” is relevantly appropriate.